# From Severe Anemia to Intestinal Hemangiomatosis, a Bumpy Road—A Case Report and Literature Review

**DOI:** 10.3390/diagnostics14030310

**Published:** 2024-01-31

**Authors:** Raluca Maria Vlad, Ruxandra Dobritoiu, Carmen Niculescu, Andreea Moga, Laura Balanescu, Daniela Pacurar

**Affiliations:** 1“Grigore Alexandrescu” Emergency Children’s Hospital, 011743 Bucharest, Romania; ruxandra.hp@gmail.com (R.D.); andreea.moga@umfcd.ro (A.M.); laura.balanescu@umfcd.ro (L.B.); daniela.pacurar@umfcd.ro (D.P.); 2Department of General Medicine, Pediatrics, “Carol Davila” University of Medicine and Pharmacy, 020021 Bucharest, Romania; 3“Sf Ioan” Children’s Hospital, 800487 Galati, Romania; pavaleanucarmen@yahoo.com

**Keywords:** chronic anemia, intestinal hemangiomas, digestive bleeding

## Abstract

Gastrointestinal hemangiomas (GIH) are unusual vascular tumors found anywhere alongside the GI tract, the small bowel being the most common site. Diagnosis requires good clinical insight and modern imaging. This is a comprehensive review of the literature, starting from a new pediatric case diagnosed through exploratory laparotomy after complex imaging techniques failed. This research was conducted on published articles from the past 25 years. We identified seventeen original papers (two series of cases with three and two patients, respectively, and fifteen case reports). The female/male ratio was 1.5. The youngest patient was a 3-week-old boy, and the was oldest a 17-year-old girl. The most common localization was the jejunum (eight cases), followed by the ileum (four), colon (three), stomach (two), and rectum (one). Seven children had cavernous and four had capillary hemangiomas. Eight patients presented gastrointestinal bleeding, seven had refractory anemia, such as our index patient, three had recurrent abdominal pain, and two had bowel obstruction. Surgical assessment was successful in fifteen cases; three cases experienced great outcomes with oral propranolol, one child was treated successfully with sirolimus, and for one patient, endoscopic treatment was the best choice. The authors present the case of a female patient admitted to the Pediatrics Department of “Grigore Alexandrescu” Emergency Children’s Hospital from 25 February to 28 March 2019 for severe anemia, refractory to oral iron treatment, and recurrent blood infusions. No clear bleeding cause had been found. Although very uncommon, intestinal hemangiomas can express puzzling, life-threatening symptoms. We should keep in mind this disorder in cases of unresponsive chronic anemia.

## 1. Introduction

### 1.1. Aim

This review and case report stand to highlight a rare and difficult-to-diagnose pediatric pathology: intestinal hemangiomas, and the literature is poor in such cases. We conducted an extensive review of the cases published in the past 25 years with a close look on our own, comparing clinical features and methods in terms of diagnostic tools and therapeutic strategies.

### 1.2. Incidence

Hemangiomas, the most common vascular tumors of infancy, are benign masses with incidences between 1% and 9%, depending on race [1]. Hemangiomas affect mostly Caucasians, with the incidence being 10–12 times higher in white children than Black or Asian pediatric populations. Girls are affected more often than boys at a 3:1 ratio [2]. Gastrointestinal (GI) involvement is uncommon, especially in the pediatric population, and is typically represented in the literature by case reports or a series of cases. It constitutes between 7 and 10% of all benign small bowel tumors and 1.6% of benign stomach tumors. Although these vascular masses may occur anywhere alongside the GI tract, the small bowel is the most common site (jejunum), followed by the large intestine (most often the recto-sigmoid) and stomach. Most of these vascular tumors are focal and solitary, but multiple lesions may occur. Also, GI hemangiomas can be associated with various syndromes, such as Maffucci’s Syndrome, Klippel–Trenaunay Syndrome, Disseminated Neonatal Hemangiomatosis, or Blue Rubber Bleb Nevus Syndrome [2,3,4,5].

### 1.3. Background

The cause and original cell of child hemangiomas have not yet been settled, but there is a strong relationship between these vascular tumors and placental micro-vessels. Some reports say that infants born from mothers who underwent chorionic villus sampling are at higher risk of developing hemangiomas. Regarding the hereditary component in the development of infantile hemangiomas, there is not enough data to support it, because most cases appear to be sporadic [2].

Cutaneous hemangiomas, commonly affecting upper body extremity, often follow an uneventful course; thus, the treatment consists of observation. Natural evolution is marked by early proliferation over the first year of life, followed by progressive spontaneous involution in 1 to 7 years. Visceral hemangiomas are relatively rare and can involve the liver, GI tract, or central nervous system. If there are five or more cutaneous hemangiomas, one should suspect the possibility of visceral tumors associated. Depending on sight, volume, and shape, visceral vascular tumors can cause life-threatening symptoms such as output cardiac failure, coagulopathies, GI hemorrhage, or bowel perforation [1,2,3].

Regarding histological traits, there are three main types of hemangiomas—capillary, cavernous, which can infiltrate large parts of the intestine/mesentery, and mixed. Histologic confirmation of hemangiomas relies on a positive stain for glucose transporter 1 (GLUT-1) [2,6].

Capillary hemangiomas are tumors consisting of branching vessels with a narrow antrum, which is not necessarily filled with blood. Macroscopic features include red or cyanotic nodes with a smooth or tuberous surface [3,7]. Cavernous hemangiomas, the most common type, can present as intraluminal polypoid tumors or diffuse submucosal and intramural irregularities. Involved blood-filled sinuses with a scarlet-cyanotic node appearance are clearly separated from the surrounding tissue (“sponge on the cut” lookalike) [7]. Capillary hemangiomas are usually isolated, causing chronic minor bleedings and manifesting as anemic syndrome. Cavernous hemangiomas can cause acute massive GI bleeding, resulting in significant hematemesis or melena [2,3].

### 1.4. Diagnostic and Therapeutic Features

Patients usually present with signs of acute or chronic GI bleeding, but complications such as obstruction, perforation, or intussusception may arise. Stojsic et al. described the case of a 13-year-old girl with ileal capillary hemangioma presenting with intestinal obstruction [8]. Singh et al. reported the case of a solitary intussuscepting capillary hemangioma in the distal ileum presenting with perforation and peritonitis [9]. Laarif et al. mentioned the case of a 13-week-old girl with ileum hemangioma who presented with intestinal intussusception [10].

Laboratory studies for hemangiomas should include the careful assessment of anemia markers, such as serum iron levels, total iron binding capacity, ferritin, and serum transferrin levels. Thus, pathologies like anemia of chronic diseases, iron deficiency anemia, sideroblastic anemia, or thalassemia should be ruled out from the start. Inflammatory bowel disease and celiac disease, such as malabsorption conditions, should also be excluded from the beginning by specific biomarkers. Unfortunately, there are no particular laboratory studies to diagnose hemangiomas; the literature reports discuss the vascular endothelial growth factor, the urinary beta-fibroblast growth factor, or matrix metalloproteinases as markers of hemangioma proliferation, but the practical use of these indices has not yet been settled [2].

Regarding imaging diagnostic tools, clinicians should first perform an abdominal ultrasound to rule out any obvious causes of bleeding, such as vascular malformations. High vessel density and high peak arterial Doppler shifts are specific for infantile hemangiomas [2]. Upper endoscopy and colonoscopy are next in line for identifying bleeding sites, although a biopsy of potential lesions is not recommended because of the significant risk of massive bleeding. Bluish-like submucosal dilated venous lesions alongside the bowel wall can be observed on a colonoscopy [11]. If there are no findings of a bleeding source, hemangiomas are suspected to be located in the small bowel; thus, wireless capsule endoscopy should be performed. Bae et al. concluded that capsule endoscopy is an adequate diagnostic tool for children with obscure chronic GI bleeding [12]. It is considered a noninvasive procedure that allows visualization of the entire small intestine but has downsides, as it cannot be used in infants because of high obstruction risk and does not permit tissue sampling [11]. “Push-and-pull” enteroscopy, which is safe and efficient in pediatric patients, could also be a great tool to evaluate small bowel [3,13], but capsule endoscopy is considered superior when it comes to clinically significant conditions (such as severe chronic GI blood loss) [12].

Barium enema contrast shows mucosal nodular filling defects, which narrow the intestinal lumen or compressible polypoid intraluminal masses when it comes to small bowel hemangiomas [6,11]. For colonic hemangiomas, barium enema identifies soft, serpentine masses and polypoid or circumferential lesions as seen in Figure 1 [6].

Abdominal X-rays may show phleboliths, a pathognomonic sign, especially in colorectal hemangiomas of young patients and gastric hemangiomas [6].

Magnetic resonance imaging (MRI) is the investigation of choice when it comes to locating the border and extent of a lesion [2]. Computed tomography (CT) shows diffuse wall thickening, exophytic masses, polypoid intraluminal masses, or mucosal irregularities as seen in Figure 2 [4,6].

Differential diagnosis of GI hemangiomas depends on localization. For small bowel lesions, benign and malignant tumors, such as lymphomas, primary peritoneal malignancies, or metastatic disease, should be taken into consideration. Colonic lesions should be differentiated from inflammatory diseases, infectious conditions, and carcinomas. When it comes to the differential diagnosis of gastric hemangiomas, metastatic disease and carcinomas must be ruled out [2,6].

Treatment may be conservatory, consisting of regular follow-ups, red blood cell infusions, or iron supplements for patients with small, single tumors and mild symptoms. Other therapies that proved their efficacy are corticosteroids (oral, topical, or intralesional) because they suppress the expression of vascular endothelial growth factor, alpha-interferon (especially in diffuse hemangiomatosis), and beta-blockers. Propranolol is well known for reducing the dimensions of infantile hemangiomas and is considered a first-line treatment for troublesome lesions. In addition, the FDA has approved a formulation of oral propranolol solution suitable for 5-month-old babies [2,3]. Sans et al. concluded that propranolol administered orally at a dosage of 2–3 mg/kg/daily had immediate effects on the color and growth of intestinal hemangiomas, with mild and limited side reactions and good clinical tolerance [14]. Siu Ying Angel et al. reported an unusual case of omental hemangiomatosis, which responded well to oral propranolol [15]. Cura-Esquivel et al. described the case of a 10-month-old girl with mesentery hemangioma treated with 3 mg/kg/day propranolol with complete involution of hemangioma after 6 months [16].

Laser therapy should be performed only for temporary symptom relief because of high-rate recurrence [2,11].

Surgical treatment may be used in children non-responsive to prior methods, with large or multiple hemangiomas as seen in Figure 3 [6] and associated comorbidities [3]. Surgical intervention consists of segmental resections if hemangiomas are detected in the small bowel. For colonic localization of lesions (Figure 4 [6], right or left hemicolectomy, ileo-cecotomy, or segmental resections are of choice. When it comes to recto-sigmoid hemangiomas, Oner and Altaca recommended low anterior resection in 1993; this practice remains relevant today because it preserves the sphincter [2,3,11].

## 2. Detailed Case Description

A girl that was 6 years and 6 months presented with severe anemia in August 2018 at “St. Joan” Children’s Hospital Galati (the lowest hemoglobin level was 2.8 g/dL). Subsequently, the patient had two other admissions with the same diagnosis. She underwent extensive investigations to establish the cause of chronic anemia. Infectious sources, hemolytic anemia, and malignancies were excluded. She required repeated blood transfusions and intravenous iron infusions. Hemoglobin values were decreasing shortly after substitutive treatment (down to hemoglobin values of 5–6 g/dL), without a personal history of active bleeding. The Adler test (occult hemorrhages in feces) was persistently positive. Compliance with oral iron treatment at home was poor. She came from a broken family (both parents had gone abroad), and she was raised by an illiterate grandmother in very low socio-economic conditions. Her pediatrician in the regional hospital brought her in for check-ups with police intervention. In February 2019, she was transferred to our department for further investigations.

On admission, the patient was associated with pallor and failure to thrive (weight = 12.5 kg; height = 105 cm; BMI = 12 kg/m^2^, z-score = −4.98) and had a systolic murmur on cardiac auscultation; she had no liver or spleen enlargement but exhibited persistent black stools (with normal consistency and no marks of fresh blood).

Drawing a relevant history from the grandmother was challenging, as she could not provide details regarding the onset of black stools that the patient had at admission.

Investigations revealed severe regenerative anemia, low iron serum levels, anisocytosis, poikilocytosis with normal leukocyte and platelet counts, no inflammatory syndrome, normal fecal calprotectin, and a positive Adler test. Hemoglobin levels during the patient’s hospital stay are illustrated in Figure 5.

Abdominal ultrasound and abdominopelvic CT scan showed normal findings. Upper and lower digestive endoscopy did not reveal signs of active bleeding and pathology on gut biopsies was unremarkable. Tc^99^ scintigraphy did not describe Meckel’s diverticulum.

Celiac disease and bowel inflammatory disease, commonly considered in such cases, were ruled out: there were normal anti-tissue transglutaminase antibodies, no suggestive findings on the upper and lower endoscopy, and normal fecal calprotectin [2]. 

Moreover, less frequent pathologies causing chronic bleeding and iron-supplement refractory anemia, like collagenous gastritis or chronic autoimmune atrophic gastritis, were disbarred on the basis of endoscopic findings and a lack of clinical and immunological features and pernicious-type anemia, respectively [17,18].

An exploratory laparoscopy was performed. The cecum was identified, and the small bowel was then checked, with multiple vascular tumors being detected alongside the small bowel loops. No such lesions were detected in the colon. Due to the large number of lesions, the surgical team decided to convert to exploratory laparotomy. Various vascular tumors were found in the ileum and jejunum, so all of them were removed and intestinal anastomoses were performed (Figure 6). A peritoneal drainage was kept in place for 24 h and was subsequently removed. The patient had good postoperative outcomes with no complications.

An anatomopathological examination confirmed the diagnosis of intestinal hemangiomatosis (Figure 7).

Two months later, the patient had an ascending weight curve, hemoglobin level = 12.1 g/dL, and a negative Adler test (Figure 8). Short-term follow-up (6 and 12 months) had excellent findings: normal values of hemoglobin and serum iron and negative Adler tests. Long-term follow-up was not possible, as the patient came from a broken family with no adequate parental support and did not return for the subsequent scheduled medical check-ups.

## 3. Discussions

This literature review is based on PubMed papers published in the last 25 years regarding GI hemangiomas with a primary interest in the pediatric field. Particular search terms in various combinations, such as “gastrointestinal hemangioma”, “bowel hemangioma”, “digestive bleeding”, “chronic anemia”, or “oral propranolol”, were used in order to find the most suitable studies for this review. The inclusion criteria were a definite diagnosis of gastrointestinal hemangioma and follow-up after medical/surgical therapy. The exclusion criteria were incomplete patient history or diagnostic work-up, non-English papers, and adult patients with gastrointestinal hemangiomas.

We identified seventeen original papers as seen in Table 1 (two series of cases with three and two patients, respectively, and fifteen case reports). 

We assessed gender and age distribution, the site of hemangioma in the gastrointestinal tract, the type of hemangioma (capillary, cavernous, or mixed), presenting symptoms, and therapy. According to the current reviewed literature, intestinal hemangioma among the pediatric population is scarce. The female/male ratio was 1.5. The youngest patient was a 3-week-old boy, and the oldest patient was a 17-year-old girl. The most common localization for hemangioma was the jejunum (eight cases), followed by the ileum (four cases), colon (three cases), stomach (two cases), and rectum (one case). Seven children presented cavernous hemangiomas, and four were identified as having capillary hemangiomas. Eight patients presented gastrointestinal bleeding, seven had refractory anemia, such as our index patient, three had recurrent abdominal pain, and two had bowel obstruction. Surgical assessment was successful in fifteen cases; three cases experienced great outcomes with oral propranolol, one patient was successfully treated with sirolimus, and for one patient, endoscopic treatment was the best choice.

GI hemangiomas are uncommon vascular tumors that can be found anywhere alongside the GI tract, such as isolated lesions or multiple tumors, and the small intestine is the most common site. There are three main types of hemangiomas—capillary, cavernous, and mixed. The pathology report described cavernous hemangiomas in our patient.

Shen et al. reported a case of a 15-year-old girl with colonic hemangioma [11,28,29]. Han et al. reported three cases of cavernous hemangioma in the jejunum/rectum/stomach (13-month-old boy, 8-year-old boy, 15-year-old girl) [3,20,30,31], and Jones et al. reported two cases (10- and 7-year-old girls) of jejunum hemangioma [19,32]. Our patient exhibited disseminated endoluminal vascular tumors in both the ileum and jejunum.

In terms of clinical features, most patients experienced bleeding (melena or/and hematochezia), resulting in severe anemia. Bleeding can be either massive and life-threatening or recurrent and undetectable, making the diagnostic process a lot more difficult. Capillary hemangiomas are usually solitary and induce minor, well-tolerated anemia by chronic bleeding. Cavernous hemangiomas cause acute excessive hemorrhage, resulting in melena and hematemesis. Because of their scarce nature, they are not usually a highly suspected cause of GI bleeding in children. In addition, various symptoms have been described: abdominal discomfort or pain, nausea, vomiting, fatigue, dizziness, and general weakness [2,3,10,16,20,21,26,30,31,33].

Shen et al. reported the case of a 15-year-old girl with colonic hemangioma referred for a 3-week history of intermittent passage of bright red blood after defecation [11,28,29]. Han et al. describe three cases of GI hemangioma admitted for melena and vomiting with a 3-month history of hematochezia and recurrent hematemesis [3,20,30,31]. Jones et al. reported two cases of jejunum hemangioma submitted for severe iron deficiency anemia and abdominal pain, respectively, with a 3-week history of lethargy and abdominal pain [19,32]. Kim et al. reported a 17-year-old girl admitted for several months of refractory anemia [22], and Akcam et al. reported a 16-year-old boy with a 2-month history of fatigue and severe anemia [23]. Our patient was admitted for severe refractory iron deficiency anemia (the lowest hemoglobin level was 2.8 g/dL) for over 6 months, pallor, and failure to thrive.

There are no universally accepted laboratory studies for the diagnosis of infantile hemangioma. Serum vascular endothelial growth factor (VEGF), urinary beta-fibroblast growth factor, and matrix metalloproteinases (MMPs) are used as markers of hemangioma proliferation and differentiation [2]. The use of glucose transporter 1 (GLUT-1) is a very sensitive and specific method for histological confirmation of infantile hemangioma; it is helpful for evaluating tissues removed during biopsy or excision. Proliferating and involuting hemangiomas stain positively for GLUT-1, while vascular neoplasms and vascular malformations do not [2,34]. None of these markers were available in the diagnostic race for our patient.

Radiologic images of phleboliths are pathognomonic in GI hemangiomas [4,6,35,36,37]. Imaging diagnostic methods rely on barium enema, abdominal X-ray and ultrasound, CT, MRI, scintigraphy with T^99^, and angiography. Upper and lower endoscopies can also be helpful tools, depending on the structural features of the bleeding site. [2,3,20,30,31,33]. A capsule endoscopy is performed when bleeding is suspected to be in the small bowel and the hemorrhage site cannot be detected by plain endoscopy. However, this method has limited use if there is an intermittent bleeding spot or if the patient is too young to swallow the capsule and has a high obstruction risk. [3,20,21,30,31,38].

In 2011, Lin and Erdman investigated double-balloon enteroscopy (DBE) in the pediatric population. They determined that DBE is safer for the diagnosis and treatment of hemangiomas, but it should be reserved for patients with high suspicion of small intestine involvement only after less-invasive techniques have failed [3].

MRI is the imaging method of choice, as it provides the location and extent of cutaneous and extracutaneous hemangiomas. It also helps exclude other high-flow vascular lesions, such as arteriovenous malformations [2,4,6,35]. For our patient, abdominal CT showed no structural anomalies, and T^99^ scintigraphy was performed to rule out Meckel’s diverticulum, which came out negative. An upper endoscopy found no bleeding evidence. A colonoscopy did not describe masses or active bleeding. Capsule endoscopy was not an option, as the patient was too young to swallow the device.

Conservative treatment for GI hemangiomas consists of oral iron supplementation and red blood cell infusions in cases of symptomatic severe anemia [3,20,30,31]. Our patient required repeated blood transfusions and intravenous iron infusions, but hemoglobin levels were decreasing shortly after substitutive treatment (down to hemoglobin values of 5–6 g/dL), without a personal recent history of active bleeding.

Therapy options include steroids (providing the suppression of vascular endothelial growth factor) or interferon-alfa, which proved to be effective for diffuse hemangiomas and anti-angiogenic agents. Requirements for steroid therapy are large hemangiomas with life-threatening hemorrhage risk [2,3,20,21,30,31]. None of this medication was administered to our patient.

In 2008, propranolol was approved for the treatment of infantile hemangiomas and as first-line therapy for liver hemangiomas. It has a down-regulation effect on vascular growth factors, suppresses GLUT-1 receptors, and induces apoptosis. Is a safe, well-tolerated drug with a long use history in pediatric cardiology, despite a lack of large clinical trials. This medication proved its supremacy over steroids, vincristine, and laser therapy in reducing the volume, color, and elevation of hemangiomas [1,9,14,16,23,39].

Rubinstein et al. reported the case of a 3-week-old boy admitted for large hemangioma of the omentum and mesentery encasing the splenic vein, celiac artery, and inferior vena cava. He received oral propranolol, and after six months, the vascular mass began to decrease in volume [1,9,14,39]. Akcam et al. described the case of a 16-year-old boy with multiple cavernous hemangiomas in the stomach and colon; he received a high dose of propranolol, and after 8 months, the tumors began to shrink [23]. Cura-Esquivel et al. reported the case of a 10-month-old girl with mesentery hemangioma with complete involution of lesions after 6 months of oral propranolol [16].

Propranolol oral solution was approved for pediatric use by the FDA in 2014. A study showed that infants treated with propranolol at a total dosage of 2 mg/kg/day were less likely to undergo invasive procedures [2,15,16]. Our patient did not receive oral propranolol because hemangioma diagnosis was established on exploratory laparotomy and during surgery, the vascular masses were removed; thus, there was no need for beta-blocker therapy.

Children unresponsive to medical care should undergo surgical evaluation and treatment. Sometimes, such as in our case, exploratory surgery is the only diagnostic key tool (AE Jones et al. support this judgment) [3,19,20,30,31,32]. There are multiple surgical procedures that can be performed, depending on the site of hemangiomas. For stomach tumors, subtotal or total gastrectomy, antrectomy, or wedge resection are available [3,20,30,31]. For small bowel and colonic hemangiomas, segmental resections are the choice [2,3,10,20,21,30,31]. Our patient underwent exploratory laparotomy, which encountered multiple ileal endoluminal vascular tumors—one purple-red lesion about 40 cm away from the ileocecal valve and three similar lesions on the ileum. The removal of each hemangioma with termino-terminal anastomoses was performed. Pathology examination of the removed pieces found a vascular-type tumor. After surgery, recovery was swift, and the patient was rapidly discharged with a complete course of oral antibiotics and oral iron supplementation. Subsequent monitoring did not reveal any decrease in the hemoglobin levels, and no blood was found in the stool for the next year. Long-term follow-up of these patients could give us an adequate idea of how monitoring should be performed post-surgery. To this day, no data regarding long-term prognosis are available and unfortunately, our case could not provide new insight on this subject, as the patient was lost after 12 months of follow up.

The limitations of this review consist primarily of the scarce literature on this topic and the very few resources spread over a long period of time, making comparison and conclusion drawing very difficult and possibly inaccurate. Future research paths may lie with vascular growth factors and matrix metalloproteinases, as they might be used as markers of hemangioma proliferation and differentiation. Also, GLUT-1 staining should be considered and further evaluated as a histological marker of infantile hemangioma.

## 4. Conclusions

GI hemangiomas are rare benign tumors primarily affecting the small bowel. They most often occur in infancy and childhood, are more frequent in girls, and may be associated with various syndromes, such as “Blue rubber bleb” or Klippel–Trenaunay Syndrome. Patients may present general weakness and dizziness and show any degree of anemia due to chronic bleeding from a capillary-type hemangioma. They may exhibit hematochezia and melena due to acute severe bleeding from a cavernous-type hemangioma. These infantile vascular tumors manifest as intramural or intraluminal masses, and the presence of phleboliths on radiologic images is pathognomonic. Conservative pathogenic treatment includes propranolol as first-line therapy, followed by steroids, anti-angiogenic agents, and interferon-alfa. Blood transfusions and oral iron supplementation should be used as supportive treatment. Patients unresponsive to medical interventions benefit from surgical treatment.

Although a rare condition in the pediatric population, intestinal hemangiomas can cause serious life-threatening symptoms, and the clinician should consider this disorder in cases of unresponsive chronic anemia.

## Figures and Tables

**Figure 1 diagnostics-14-00310-f001:**
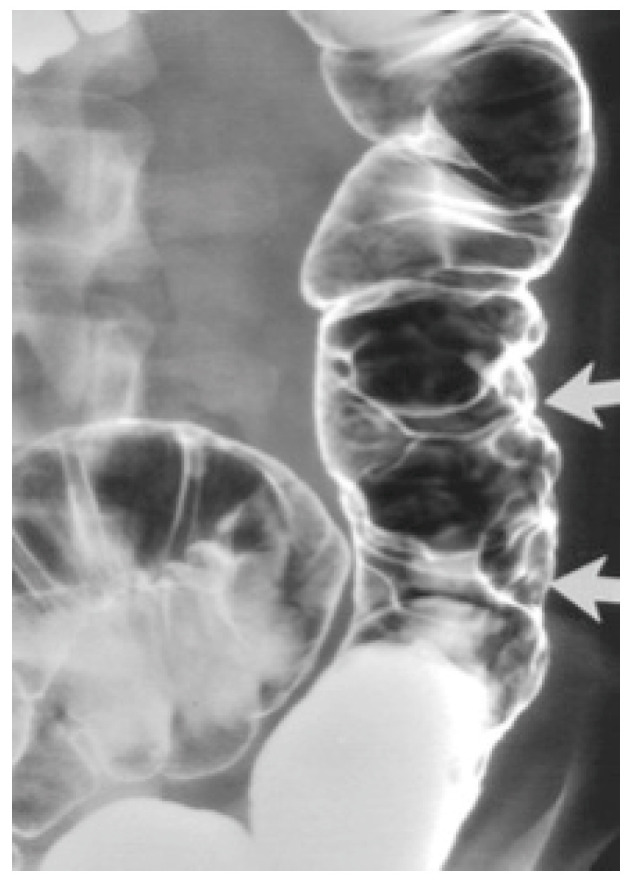
Cavernous hemangioma of a descending colon—air-contrast barium enema [6].

**Figure 2 diagnostics-14-00310-f002:**
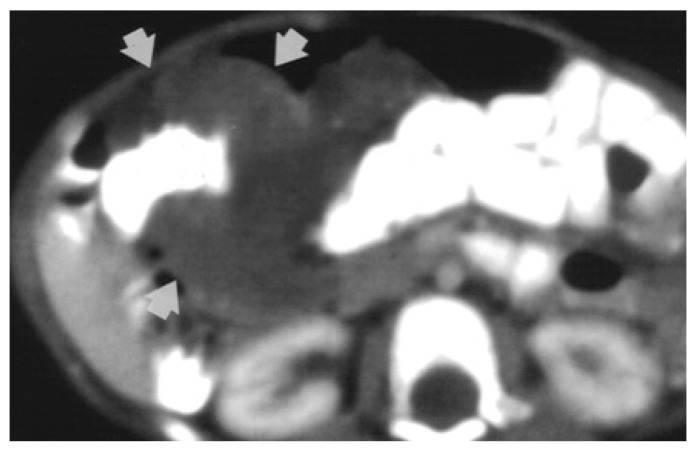
CT-scan shows circumferential infiltration of the colon by soft-tissue attenuation mass (9-month-old girl with cavernous hemangioma of the colon) [6].

**Figure 3 diagnostics-14-00310-f003:**
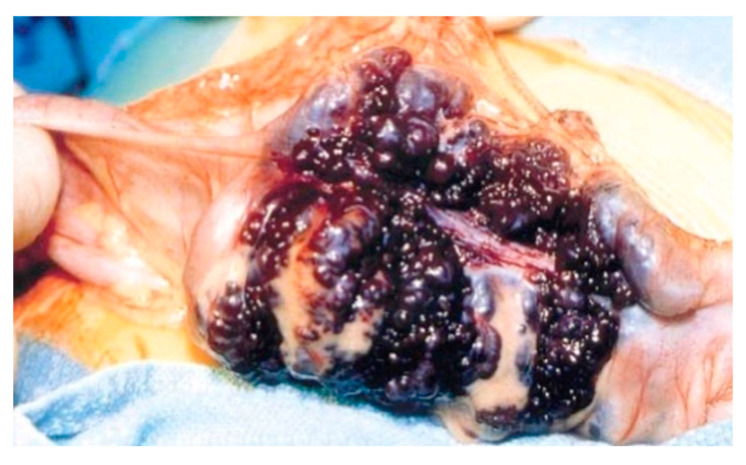
Intraoperative image showing circumferential mass composed of large blood-filled spaces [6].

**Figure 4 diagnostics-14-00310-f004:**
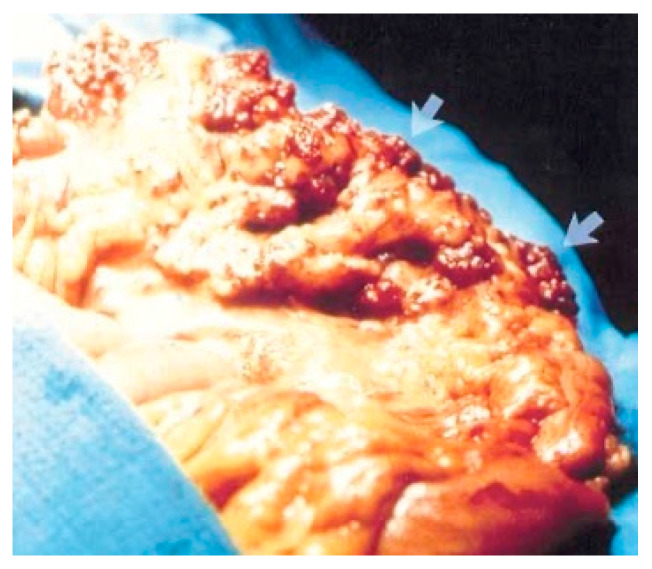
Cavernous hemangioma of a descending colon—photographed during surgery [6].

**Figure 5 diagnostics-14-00310-f005:**
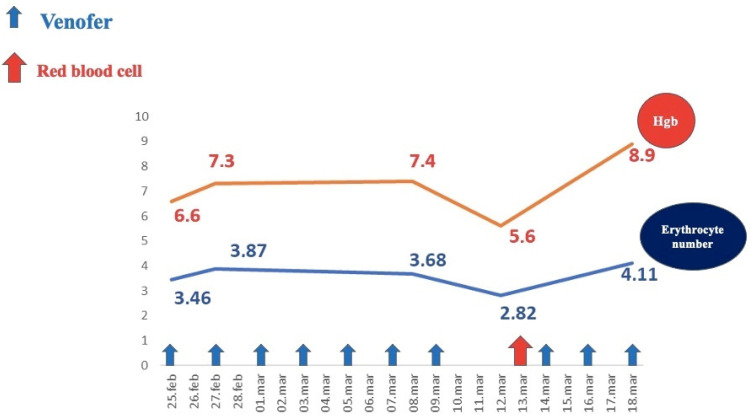
Evolution of hemoglobin levels under medical treatment.

**Figure 6 diagnostics-14-00310-f006:**
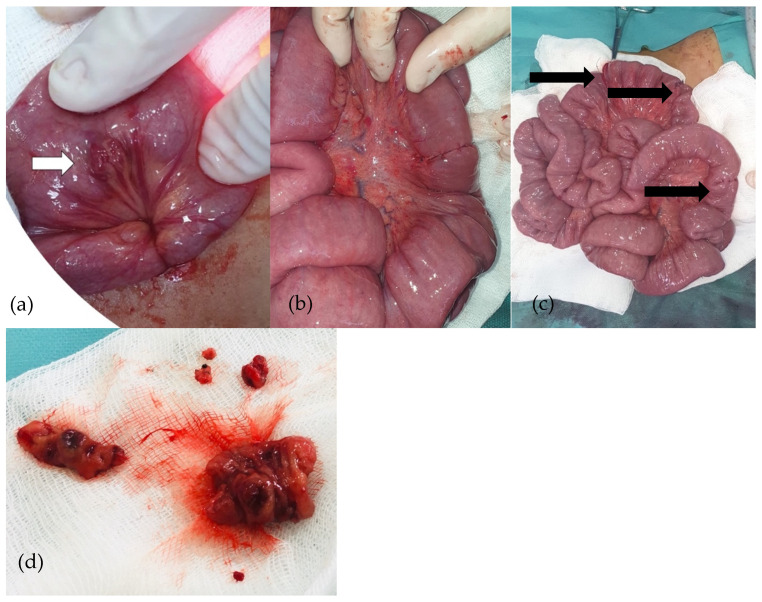
(**a**–**d**): Purple-red lesion on the ileum (**a**). Cuneiform resections and ileo-ileal anastomoses (**b**). Ileo-ileal anastomoses (**c**). Removed ileal piece (**d**).

**Figure 7 diagnostics-14-00310-f007:**
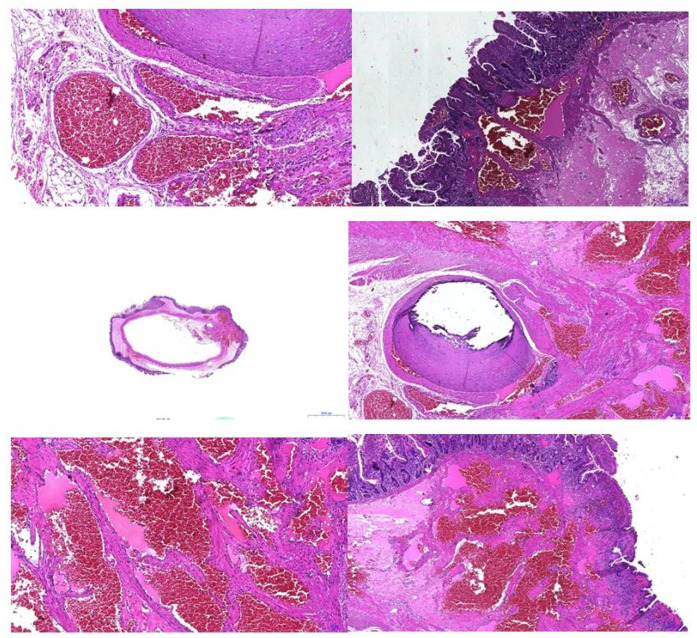
Pathologic examination—vascular anomaly.

**Figure 8 diagnostics-14-00310-f008:**
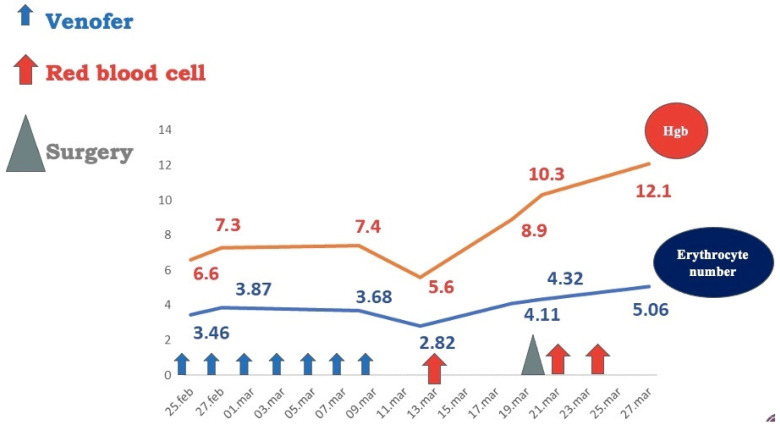
Patient hematological output after surgical removal of vascular tumors.

**Table 1 diagnostics-14-00310-t001:** Literature cases of GI hemangiomas.

Article by Year	Subject Age	Site and Type of Hemangioma	Presenting Symptoms	Therapy Intervention and Follow-Up
Han et al., 2014 [3]	13 months/male	Jejunum/Cavernous hemangioma involving submucosa to the subserosa	Melena and nonbilious vomiting for 15 days	Resection/discharged on the seventh day and no further issues
	8 years/male	Rectum/Cavernous hemangioma	Hematochezia for 3 months	Resection/discharged on the twelfth day and no further issues
	15 years/female	Jejunum, distal ileum/cavernous hemangioma	Dizziness	Resection/Recurrence of multiple hemangiomas in the stomach and small bowel 4 years later; no anemia or intestinal bleeding
Rubinstein et al., 2014 [1]	3 weeks/male	Omentum and mesentery/type not specified	Intra-abdominal hemorrhage	Oral propranolol
Karaseva et al., 2018 [7]	11 years/male	Jejunum/Cavernous hemangioma	Chronic anemia and recurrent intestinal bleeding	Resection and end-to-end anastomosis
Jones et al., 2007 [19]	1 year/female	Jejunum hemangioma/type not specified	Severe iron deficiency anemia and abdominal pain	Resection and end-to-end anastomosis Uneventful recovery No anemia at 6 months follow-up
	7 years/female	Jejunum hemangioma/type not specified	A 3-week history of lethargy and abdominal pain	Resection and end-to-end anastomosis Uneventful recovery No anemia at 6 months follow-up
Shen et al., 2016 [11]	15 years/female	Colonic hemangioma involving mucosa, submucosa, pericolic fat/type not specified	A 3-week history of the intermittent passage of bright red blood after defecation	Anterior resection and end-to-end anastomosis Full recovery No further bleeding
Kavin et al., 2006 [20]	2.5 years/female	Jejunum/Mixed capillary hemangioma	Melenic stools	Resection/no further issues
Coleman et al., 2018 [21]	2 years/female	Small bowel hemangioma/type not specified	Severe microcytic anemia and melenic stools	Resection of lesion/no further issues
Kim et al., 2004 [22]	17 years/female	Diffuse hemangiomatosis, largely distributed in the stomach and colon/type not specified	Several months of refractory anemia and dizziness	Endoscopic ligation Improved anemia
Akcam et al., 2015 [23]	16 years/male	Multiple cavernous hemangiomas in the stomach and colon	A 2-month history of fatigue and severe anemia	Oral propranolol 2 mg/kg/day After 8 months, hemangiomas volume significantly decreased
Bae et al., 2015 [12]	13 years/male	Jejunum/Cavernous hemangioma	Recurrent iron-deficiency anemia, nausea, and dizziness	Resection/discharged after 7 days and no further issues
Turcotte et al., 2012 [24]	16 years/female	Jejunum/Capillary hemangioma	Fatigue and weakness	Resection/no further issues and uneventful recovery
Stojsic et al., 2008 [8]	13 years/female	Ileum/Capillary hemangioma, pyogenic	Intestinal obstruction	Resection/no further issues
Sakaguchi et al., 1998 [25]	11 years/male	Ileum/Cavernous hemangioma	Recurrent anemia and abdominal pain	Resection/discharged after 7 days and no further issues.
Cura-Esquivel et al., 2021 [16]	10 months/female	Mesentery hemangioma/type not specified	Hematochezia for 3 months and anemia	Oral propranolol 3 mg/kg/day After 6 months, hemangioma disappeared
Laarif et al., 2023 [10]	13 weeks/female	Ileum hemangioma/type not specified	Bilious vomiting, rectal bleeding for 3 days and intestinal intussusception	Segmental ileal resection Good outcome at 6 months
Kumar et al., 2022 [26]	4 months/male	Abdominal involuting hemangioma (liver surface)/capillary hemangioma	Bilious vomiting for 2 days and partial bowel obstruction	Resection of involuting lesion adhered to the liver Good outcome at 2 years
Kleinman et al., 2023 [27]	3 months/female	Diffuse small bowel hemangioma (PHACE syndrome)/type not specified	PHACE syndrome symptoms (with cardiovascular anomalies, hemangioma)—limited access article	Oral propranolol and sirolimus (GI symptoms persisted under oral propranolol) The addition of sirolimus led to regression of hemangioma (first reported case)

## Data Availability

Data are contained within the article.

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
