# Peer review of "From Severe Anemia to Intestinal Hemangiomatosis, a Bumpy Road—A Case Report and Literature Review"

_diagnostics, 2024, doi:10.3390/diagnostics14030310_

Round 1

Reviewer 1 Report

Comments and Suggestions for Authors

I have several major comments for revision:
1. The Introduction lacks a clear statement of the study's objective or research question. Specify the aim of the study early on to guide readers. Also, consider breaking down the Introduction into subsections, such as background, incidence, and clinical manifestations, for better organization.
2. Some statements lack proper citations, like the incidence of hemangiomas. Ensure that credible references support all statistics and facts.
3. There is repetition of information throughout the text. Streamline the content by eliminating redundant details and emphasizing key points. For example, the information about hemoglobin levels is repeated, which could be condensed for clarity.
4. Clarify ambiguous terms, such as "Addler test". Provide a brief explanation or replace it with a more widely recognized term.
5. The text contains grammatical errors and awkward phrasing. Edit sentences for clarity and coherence. For instance, in line 113, revise phrases like "althought biopsy of eventual lessions is not reccomended" to "although biopsy of potential lesions is not recommended".
6. The discussion section could benefit from a more in-depth analysis of the findings. Connect the presented case with the broader literature and discuss implications for clinical practice. Also, consider discussing the limitations of the study and proposing areas for future research to enhance the scholarly value.

Comments on the Quality of English Language

The text contains grammatical errors and awkward phrasing. Edit sentences for clarity and coherence. For instance, in line 113, revise phrases like "althought biopsy of eventual lessions is not reccomended" to "although biopsy of potential lesions is not recommended".

Reviewer 2 Report

Comments and Suggestions for Authors

The article "From Severe Anemia to Intestinal Hemangiomatosis, a Bumpy Road – A Case Report and Literature Review," details the medical journey of a 6-year-old girl with severe, chronic anemia, presenting a complex diagnostic and therapeutic challenge. Despite extensive investigations, including infectious disease screening, hemolytic anemia, malignancies, and various imaging techniques, the cause of her anemia remained elusive until an exploratory laparotomy revealed multiple vascular tumors in the ileum and jejunum. This surgical intervention, involving the removal of the tumors and subsequent intestinal anastomosis, led to a successful outcome with the patient showing significant clinical improvement.

The literature review encompassed 17 papers focusing on gastrointestinal hemangiomas in the pediatric population, offering insights into the rarity and variability of this condition.

Areas for Improvement

- To enhance the comprehensiveness of a more complete differential diagnosis should be introduced, especially focusing on the major causes of chronic anemia in pediatric population. Specifically, when the article mentions, "Upper (figure 8) and lower (figure 9) digestive endoscopy did not reveal signs of active bleeding and pathology on gut biopsies was unremarkable," it would be beneficial to explicitly state that celiac disease and inflammatory bowel diseases (IBD), commonly considered in such cases, were excluded. Additionally, the exclusion of less frequently considered conditions, such as microscopic (collagenous) gastritis and chronic autoimmune gastritis, should be mentioned. References to relevant PubMed IDs (PMID 38034434) would provide valuable supporting evidence for these exclusions.

Moreover, the article should clarify that the patient did not have recurrent infections from COVID-19, which can cause bleeding and may impact anemia. This aspect is particularly relevant given the ongoing global health context and the emerging literature on COVID-19's systemic effects, including gastrointestinal manifestations (referenced with appropriate PMIDs: 32480008).

Furthermore, the article would benefit from removing images of tests that yielded normal results, particularly when these do not add significant value to the understanding of the case. Specifically, the ultrasound, endoscopic, and CT images that did not contribute to the diagnosis could be omitted to streamline the focus on the more pertinent findings and interventions that led to the diagnosis of intestinal hemangiomatosis. This approach would enhance the clarity and focus of the article on the critical aspects of the case.

-A more extensive discussion on the potential long-term follow-up and management of such cases post-surgery could be beneficial.

Comments on the Quality of English Language

Generally good

Author Response

Answer paragraph 1: I addressed and completed all the issues regarding a more extensive differential diagnosis.

Answer paragraph 2: Covid-19 testing was not performed as the patient presented to our department in 2018, before the pandemic.

Answer paragraph 3: I have deleted the figures (ultrasound, endoscopic and CT) that showed normal findings and did not significantly contribute to the positive diagnosis.

Answer paragraph 4: I added details on the patient follow-up and also discussed the importance and extension of long-term follow-up post-surgery for these cases.

Round 2

Reviewer 1 Report

Comments and Suggestions for Authors

Thanks for your revisions.

Author Response

  1. I broke down the Introduction area into 3 fields: Incidence, Background, Diagnostic and Therapeutic features. At the beginning, I have written a clear statement regarding the aim of this paper. I have also moved some paragraphs in order to better fit the 3 new fields of the “Introduction".
  2. I have provided citation for that information.
  3. I deleted unnecessary information about hemoglobin levels.
  4. I clarified the term “Adler test” providing information about this procedure between brackets.
  5. The necessary corrections on English grammar and phrasing were done. Also the text was read and corrections were made by a Native English speaker.
  6. I detailed and discussed the case comparing it to the small number of case repots found in the literature. As one of few, this case might help raise the awareness of clinicians on a rare and extremely difficult to diagnose pathology. I discussed the limitations of the review and proposed a few paths for future research

Reviewer 2 Report

Comments and Suggestions for Authors

Upon re-reviewing the revised manuscript, I noticed a couple of areas where my previous suggestions have not been fully incorporated, particularly concerning the inclusion of certain key references. I believe that addressing these points could further enhance the manuscript's value and comprehensiveness.     

While I appreciate the addition of sentences regarding the exclusion of celiac disease, IBD, microscopic gastritis, and chronic autoimmune gastritis, the inclusion of the suggested references (I write the extensive citation format:

- Romano M, Plott N, Galligan A, Khalaf R. Literature Review and a Relevant Case of Pediatric Collagenous Gastritis: A Rare but Important Etiology of Iron-Deficiency Anemia. JPGN Rep. 2023 Sep 8;4(4):e351. doi: 10.1097/PG9.0000000000000351. PMID: 38034434; PMCID: PMC10684157.

- Lenti MV, Lahner E, Bergamaschi G, Miceli E, Conti L, Massironi S, Cococcia S, Zilli A, Caprioli F, Vecchi M, Maiero S, Cannizzaro R, Corazza GR, Annibale B, Di Sabatino A. Cell Blood Count Alterations and Patterns of Anaemia in Autoimmune Atrophic Gastritis at Diagnosis: A Multicentre Study. J Clin Med. 2019 Nov 15;8(11):1992. doi: 10.3390/jcm8111992. PMID: 31731715; PMCID: PMC6912578.

This would greatly strengthen this section. These references provide critical evidence supporting these exclusions and would help validate the study’s findings.

Comments on the Quality of English Language

Quite good

Author Response

I introduced the 2 references about collagenous gastritis and chronic autoimmune gastritis you kindly suggested after the relevant paragraphs (the reference numbering was also modified accordingly). 

I revised the english quality of the manuscript and tried to improve it by changing a few terms ("hyposideremia" -> low serum iron levels, "a 10 years-old" -> a 10 year-old, etc.)

Round 3

Reviewer 2 Report

Comments and Suggestions for Authors

The last revision has significantly enhanced the clarity, depth, and scientific rigor of the paper.